# Knowledge about Disease, Medication Therapy, and Related Medication Adherence Levels among Patients with Hypertension

**DOI:** 10.3390/medicina55110715

**Published:** 2019-10-28

**Authors:** Anna Gavrilova, Dace Bandere, Ieva Rutkovska, Dins Šmits, Baiba Mauriņa, Elita Poplavska, Inga Urtāne

**Affiliations:** 1Department of Pharmaceutical Chemistry, Faculty of Pharmacy, Rīga Stradiņš University, LV-1007 Riga, Latvia; Dace.Bandere@rsu.lv (D.B.); Inga.Urtane@rsu.lv (I.U.); 2Department of Dosage Form Technology, Faculty of Pharmacy, Rīga Stradiņš University, LV-1007 Riga, Latvia; Ieva.Rutkovska@rsu.lv (I.R.); dzs14464@me.com (D.Š.); Baiba.Maurina@rsu.lv (B.M.); Elita.Poplavska@rsu.lv (E.P.); 3Department of Public Health and Epidemiology, Faculty of Public Health and Welfare, Rīga Stradiņš University, LV-1010 Riga, Latvia; 4Institute of Public Health, Rīga Stradiņš University, LV-1046 Riga, Latvia

**Keywords:** medication adherence, hypertension, questionnaire, Morisky Medication Adherence Scale

## Abstract

*Background and Objectives:* A particular problem in cardiology is poor adherence to pharmacological treatment among patients with hypertension. It is known that approximately half of these patients do not use their medications as prescribed by their doctor. Patients may choose not to follow the doctor’s recommendations and regularly do not control their blood pressure, owing to many factors. A convenient method for measuring the level of adherence is the Morisky Medication Adherence Scale, which also provides insight into possible remedies for low adherence. We investigated their therapy, knowledge about the disease and its control, and demographic differences to assess the adherence of patients with hypertension. *Materials and Methods:* This was a cross-sectional observational study. Data were collected through a survey of 12 pharmacies in Latvia. The study involved 187 participants with hypertension. *Results:* The prevalence of non-adherence was 46.20% in Latvia. The oldest patients were the most adherent (*p* = 0.001, β = 0.27). The higher the self-rated extent from 0 to 10, to which the patient takes their antihypertensives exactly as instructed by their physician, the higher the level of adherence (*p* < 0.0001, β = 0.38), where at “0”, the patient does not follow physician instructions at all, and at “10”, the patient completely follows the physician’s instructions. Non-adherent patients tend to assess their medication-taking behavior more critically than adherent patients. The longer the patient is known to suffer from hypertension, the more adherent he or she is (*p* = 0.014, β = 0.19). *Conclusions:* Medication non-adherence among patients with hypertension is high in Latvia. Further investigations are needed to better understand the reasons for this and to establish interventions for improving patient outcomes.

## 1. Introduction

Hypertension is the most common cardiovascular disease in the adult population [1]. Blood pressure is recognized as a universal premorbid factor that is associated with many risk factors of various chronic diseases [2]. The Global Burden of Disease study concluded that hypertension is the key factor for disability-adjusted life years worldwide [3]. According to the World Health Organization (WHO), hypertension affects 1.13 billion people around the world (~20% of women and ~24% of men) [1]. The highest rates of men with arterial hypertension (AH) in 2015 were found in Croatia, Latvia, Lithuania, Hungary, and Slovenia, and those of women were found in Niger, Chad, Mali, Burkina Faso, and Somalia. Nearly three in ten people had AH in these countries [4].

As reported by WHO, hypertension is a cardiovascular disease that is associated with an overall increased systolic blood pressure of ≥140 mmHg and/or diastolic blood pressure of ≥90 mmHg [5]. Each 20/10 mmHg increase in blood pressure doubles the risk of cardiovascular events among certain age groups [6]. The aim of AH therapy is to achieve optimal blood pressure and reduce cardiovascular events [2]. There are also non-pharmacological recommendations, which include lifestyle changes, healthy diet, exercises, reduced alcohol consumption, and smoking cessation [5].

Pharmacological treatment includes the continuous use of antihypertensive medicine. The main drug groups that lower high blood pressure include angiotensin-converting-enzyme inhibitors, angiotensin II receptor blockers, calcium channel blockers, and diuretic agents, which are known to reduce complications and mortality among AH patients [5,7,8].

In modern cardiology practice, one of the main problems is poor adherence to pharmacological treatment [7,8]. In numerous cases, it is not possible to obtain a higher therapeutic benefit, because approximately 50% of the patients do not use their medications as prescribed by their doctor [9].

Patients may choose not to follow the doctor’s recommendations, which leads to uncontrolled blood pressure and disrupted drug therapy. This results in reduced clinical outcomes of therapy and increased cardiovascular risk [7,8]. Three-quarters of patients with hypertension do not reach an optimal blood pressure level due to poor medical adherence [3].

There are multiple factors that influence a patient’s medication adherence level, such as disease type and its severity, patient characteristics, socioeconomic factors, or the type of treatment and its regimen; there are also factors that are related to health care providers [9,10,11]. Another important factor that contributes to poor adherence in patients with AH is the asymptomatic and lifelong nature of the disease. The identification of predictors of adherence is important in reducing the risks of non-adherence in future [12].

Misinformation among patients on the proper use of the drug, recommended diet, and disease monitoring might lead to non-adherence [10,11]. Research suggests that educated patients with a high level of responsibility pay more attention to their health and are able to better manage their therapies. The patients’ active involvement in decisions regarding their own health might improve their adherence [13].

Adherence is crucial for successful antihypertensive medication therapy and blood pressure control [3]. Determining the level of adherence causes difficulties for health care specialists. Various methods, such as patient questionnaires, pill counts, assessment of patient clinical outcome, and electric medication monitoring systems, have been used for this purpose. Each method has its own pros and cons. However, patient questionnaires are most frequently used due to their simplicity and low cost, even though they are susceptible to response bias, in which participants may overstate their actual adherence. A simple method for determining the level of adherence is the Morisky Medication Adherence Scale (MMAS-8) [3,14]. It focuses on medication-taking behaviors, especially those that are related to underuse, such as forgetfulness, so that barriers to adherence can be more clearly identified. As a result, it is probably the most widely accepted self-reporting measure for adherence to medication [15].

The aim of this study was to assess patients’ adherence to medication therapy and knowledge regarding their disease in relation to their demographic characteristics.

## 2. Materials and Methods

### 2.1. Study Subjects

This was a cross-sectional, observational, quantitative, descriptive study that was conducted in primary care settings. All data were collected through face-to-face survey at 12 community pharmacies in Latvia during the period from 1 September 2018 to 1 April 2019. The selection criteria were respondent age over 18 years and diagnosis of arterial hypertension. The study involved 187 participants with hypertension—133 women and 54 men—all of whom had been taking antihypertensive medicines for at least one year. Ethical approval for this study was obtained from the Riga Stradins University Ethics Committee in 18 December 2014. Participation in the study was voluntary and anonymous. The data were collected and processed in accordance with the EU regulation 2016/679 on the protection of natural persons with regard to the processing of personal data.

### 2.2. Questionnaire

Thirty-nine questions regarding the patient’s (1) demographic information, (2) knowledge about disease and its control, (3) therapy characteristics, and (4) adherence level were included in this survey. The questions were selected based on the factors of non-adherence from the literature and previously performed studies [11,16,17,18,19,20]. Prepared questions were asked and filled in by a previously instructed pharmacist. The survey took an average of 20–30 min to complete. The questionnaire was available in the Latvian and Russian languages.

#### 2.2.1. Demographic Characteristics

This survey included information about age, gender (female or male), marital status (living alone or living with someone), education level (primary, secondary, vocational secondary, or tertiary), residence (countryside or city), employment status (unemployed, unemployed senior, employed senior, or employed), monthly income after taxes in euros (EUR) (<300, 300–600, 600–900, or >900), body mass index (BMI) (normal, overweight, or obese), smoking status (non-smokers or smokers), and physical activity (<150 or >150 min per week).

#### 2.2.2. Knowledge about Disease and Its Control

The respondents were asked questions about the optimal blood pressure level, the recommended diet, and doubts about their therapy. Additionally, it was asked whether the respondents have a blood pressure meter at home, how often they measure their blood pressure, and whether the patients think that their blood pressure is controlled.

#### 2.2.3. Medication Therapy Overview

The information about the therapy included the following aspects: how many and what kind of medications the patient uses on a daily basis (their regimen); which medications are used to treat hypertension; the use of nutritional supplements; duration of hypertension diagnosis; and, the role of the physician and/or the pharmacist in explaining the disease and medicines used. Additionally, it was asked how patients would characterize their current health status and whether they inform their physician about the nutritional supplements, non-prescription medicines, or medications that are prescribed by another specialist that they are taking. The patients had to evaluate to what extent they comply with their physician’s prescriptions on the use of medications.

#### 2.2.4. Adherence Level

The participants’ medication adherence level was assessed while using the Modified eight-point Morisky Scale (MMAS-8). MMAS-8 is a validated survey method with high reliability and validity used to evaluate the level of adherence to medication for chronic diseases, like hypertension [14,21,22,23]. We obtained permission to use this instrument in the study. The MMAS-8 consists of eight items, the first seven of which are yes/no questions, and the last of which is a five-point Likert-scale rating [14,22,23,24]. The patients can be classified and divided into three groups according to the results obtained: 8 points, 6–8 points, and 0–6 points refer to high, medium, and low levels of adherence, respectively [25]. In this study, the patients were divided into two groups—adherent and non-adherent—using a six-point “cutpoint” [16]. After completing the questionnaire, patients with fewer than six points were informed that they are non-adherent.

### 2.3. Statistical Analysis

Statistical Package for the Social Sciences (IBM SPSS Statistics 23.0; https://www.ibm.com/products/spss-statistics) was used for data analyses. Descriptive statistics were used to measure and calculate frequencies, percentages, means, and standard deviations of the sample. Non-parametric methods were employed due to the fact that the data obtained did not correspond to a normal distribution. We used the Spearman correlation to evaluate the relationships between different variables. For group comparison, the Mann–Whitney test and chi-square tests were used. All of the statistical tests were two-sided using a significance level of 0.05.

Missing data that were not related to the validated MMAS-8 method were coded as “missing value”. If there were not enough data for determining the level of adherence (i.e., not all questions were answered), the patient was excluded from the sample.

## 3. Results

Out of 187 respondents, 171 completed the full questionnaire. The average level of adherence was 5.85 (SD = 1.68): the adherent group consisted of 92 respondents, while the non-adherent consisted of 79 respondents. The prevalence of non-adherence was 46.20%.

### 3.1. Demographic Characteristics

The average age of the respondents was 64.36 (SD = 13.38); 128 of them were female (74.9%) and 43 were male (25.1%). Among these, 69.6% lived together with someone and the same percentage lived in the city. The majority of patients (44.4%) had higher education, 31.6% had vocational secondary education, 18.1% had only secondary education, and 4.7% had only primary education. According to employment, 46.2% were economically inactive retired pensioners, 38.0% were employed/self-employed, 14.0% were working pensioners, and only 1.2% were unemployed. Almost half of the respondents’ (50.9%) had a monthly income after taxes reaching EUR 300–600, 19.9% had less than EUR 300, 17.5% had EUR 600–900, and only 10.5% received over EUR 900. The weight of 21.6% of the respondents was within the permissible BMI normal, 34.5% were overweight, and 41.2% of respondents suffered from obesity. The majority (83.6%) did not smoke, while 61.4% performed physical activities for less than 150 min. per week. Table 1 and Figure 1 and Figure 2 show these parameters across adherent and non-adherent patients.

### 3.2. Knowledge about Disease and Its Control

The majority (61.4%) of respondents did not express any concerns about their medicines; the rest were not certain whether the medications are necessary, safe, and effective, or whether the illness is dangerous enough to take them. The majority (80.1%) knew about the recommended diet in the case of hypertension, but only 32.3% maintained it.

According to the answers that were obtained regarding patients’ understanding of their optimal blood pressure level, the average healthy level of systolic blood pressure should be 123.90 (SD = 6.53) mmHg. Of the respondents, 15.2% did not have a blood pressure meter at home. Nevertheless, 25.1% measured their blood pressure daily. Only 43.9% were certain that their blood pressure is controlled, while the rest admitted that it was difficult for them to understand what “controlled blood pressure” means, or they could not begin to control it systematically due to a lack of information. The frequently mentioned reasons for interrupting or terminating the therapy were forgetfulness (29.2%), concerns about getting addicted to the medications (13.5%), and undesirable side effects (10.5%).

### 3.3. Medication Therapy Overview

From the data obtained, it was concluded that 10.5% of the participants were sure that they had nothing to worry about relating to their health, while the others considered that they currently have minor, moderate, or serious health problems. The respondents used 3.92 (SD = 2.26) medications on average, of which 1.75 (SD = 0.87) were for hypertension treatment and 1.03 (SD = 1.26) were nutritional supplements. The most frequently used antihypertensive medications were angiotensin-converting-enzyme inhibitors (59.6%), β-adrenoceptor blockers (49.1%) and calcium channel blockers (48.5%), uncommon diuretics (36.8%), and angiotensin II receptor blockers (17.0%). Figure 3 provides the proportions of AH drug groups out of the total used antihypertensive medications. The duration of hypertension was 9.78 (SD = 8.09) years on average. Patients diagnosed with hypertension took their medicine once a day (46.8%). The physician (89.5%) and pharmacist (93.6%) both frequently explained the use of prescribed medications. Not all patients informed their family doctor about other medications and/or nutritional supplements that they were taking: 14.6% did not tell about medications prescribed by another physician/medical specialist, 50.3% did not speak about non-prescription medications, and 39.2% did not inform about nutritional supplements. The rest either informed their family doctor or have not used other medication and/or nutritional supplements. The respondents were asked to rate on a scale from 0 to 10 points the extent to which they follow the medication prescriptions of their family physician. The average result was 8.24 (SD = 1.76).

### 3.4. Difference between Adherent and Non-Adherent Groups

Adherent patients rated themselves higher (mean = 8.66, SD = 1.41) than the non-adherent patients (mean = 7.52, SD = 1.93) according to how much they follow the prescriptions. The higher the patient’s self-rating, the higher their level of adherence (*p* < 0.0001, β = 0.35). Non-adherent patients assessed themselves more critically than the adherent ones. The oldest patients were the most adherent ones (*p* = 0.001, β = 0.26), but the difference between genders related to adherence was not statistically significant. The average age in the adherent group was 67.73 (SD = 12,27), while that in the non-adherent group was 60.52 (SD = 13.63). The longer the patient was known to suffer from hypertension, the more adherent he or she was (*p* = 0.012, β = 0.19). Table 2 presents the results.

## 4. Discussion

The average level of adherence was 5.85 (SD = 1.68), which, according both to the eight-item self-report Morisky Medication Adherence Scale and to our own “cutpoint”, corresponds to a low level of adherence, or, in other words, to non-adherence. This result, which states that a considerable number of patients suffering from hypertension have a low level of adherence, is close to the results that were obtained in prior research [3,16,19,26,27]. This means that adherence is still an issue in cardiology.

The majority of the patients suffering from hypertension admit that they have health problems. Most of them knew about the recommended diet, but only one-third maintain it. Only one patient out of five had a BMI in the normal range. More than one-half of the respondents do not perform enough physical activity (<150 min per week) [28]. Health promotion programs for self-care health behaviors are necessary [29]. Some research has shown that maintaining a healthy lifestyle helps to reduce blood pressure. Today, most of the problems that are related to non-pharmacological treatment are similar to those that are related to the use of medications for hypertension. This situation requires further examination [12].

The termination of therapy most frequently occurs when the patients forget to take their medications or are worried about undesirable side effects [30]. However, forgetfulness, a complicated drug regimen, and side effects are also potential reasons for low adherence [17,19]. The data that we have collected correspond well to other research data; however, it turns out that every tenth patient is also worried about getting addicted to the medications [30]. Each patient is taking roughly four types of medication, two of which are specifically prescribed for hypertension and one that is a nutritional supplement. While 14.6% do not tell their physician about medications that are prescribed by another physician/medical specialist, more than one-half (50.3%) do not speak of non-prescription medications, and 39.2% do not inform about nutritional supplements. Not informing one’s healthcare specialist about any kind of medication or nutritional supplement before the therapy is assigned might have the following consequence: the patient might be at risk of double medications and mutual influence of drugs as he or she starts using several medications from the same pharmacological group. A good relationship between the patient and his or her family physician, continuous health survey, and revaluation of the therapy influence adherence positively [12,31].

E-health was introduced in Latvia. Its aim is to provide an overview of a patient’s personal data (the patient’s contact information, the family physician’s contact information, and data related to the European Health Insurance Card), basic information about the patient’s heath (diagnoses, allergies, regularly used medications, and/or medical devices), and the circulation of electronic prescriptions (the drug compensation, status—prescribed, provided partly or fully) in order to control medications and store information. E-consultations ensure that a medical specialist can provide consultations to patients as well as consulting with other medical specialists [32]. Unfortunately, there are still deficiencies in the system, but amendments are made every year. The introduction of E-health is aimed at making the communication between healthcare specialists and their patients more convenient and their collaboration more interactive and effective.

In a number of cases, the reasons for non-adherence may be related to communication problems between healthcare professionals instead of the patient’s actions. While prescribing medicine, only 74% of doctors indicate the correct name of the medical product, 87% state the indication of the prescribed medicine, 58% specify the frequency of use, and 34% indicate the duration of the treatment. This contributes to misunderstandings regarding pharmacological therapy [33]. It is necessary to improve communication based on the available data. Based on other studies, adherence to medicine can be improved by involving not only doctors, but also pharmacists [31,34,35]. In this study, we did not assess the role of the pharmacist in increasing adherence in patients with hypertension; nevertheless, it could be taken into account in future studies.

Another important issue is controlling the blood pressure. Some of the patients found it difficult to answer the question as to whether their blood pressure is controlled, because they are not aware what the control of blood pressure means. Only 43.9% of respondents were certain that their blood pressure is controlled. However, a higher proportion of medication nonadherence was noticed in the uncontrolled blood pressure patients [3,17,18,36]. Poorly controlled or untreated hypertension provokes gradual and irretrievable damage to internal organs, the consequences of which include serious complications or even death. It is important to know whether the patient understands what the norm is and how easy it is for him or her to achieve blood pressure, which is under 140/90 mmHg. Patients need the support of healthcare specialists and information that would make them understand the necessity of daily blood pressure control [5].

It is necessary to find out how to determine non-adherence easily and precisely while using the currently available methods. For this purpose, MMAS-8 can be used: it is a safe tool for detecting the adherence of those patients who suffer from cardiovascular diseases [22]. Within the scope of our research, we found a statistically significant positive correlation between MMAS-8 and the question “To what extent do you follow your doctor’s prescriptions?” In the future, it would be interesting to more thoroughly examine this correlation. Usually, patients are reluctant to fill in long questionnaires; this is why this question could be used as a precursor for researching non-adherence. If patients rate their adherence below a certain level, it will be cause for an additional questionnaire about failure to follow medical prescriptions.

A variety of determinants, both rational and emotional, which influence health behavior have been described. Adherence to pharmaceutical and non-pharmaceutical treatment is one of these behaviors. Using a conceptual framework, such as the Theory of Planned Behavior, might help in providing a deeper understanding of the findings of this research and could provide for further research initiatives [37].

In some prior research, it was found that, for every 10-year increase in age, there was a 1% absolute increase in adherence [38]. In our research, the result that the oldest patients are the most adherent correlates with this finding. However, it does not mean that there are no non-adherent patients among the oldest population. In this study, no statistically significant correlation between gender, education level, and adherence level was found, but the findings of other researchers show that women are more adherent than men and higher education contributes to patient adherence with both pharmacological and non-pharmacological antihypertensive therapy [3,16,39]. It was detected that the longer the duration of the disease, the more adherent the patient. As reported earlier, the patient is more adherent if the duration of treatment exceeds 10 years; however, there are data that show quite the opposite [16,27,39].

In this research, it was not possible to avoid the healthy adherer effect [38]. Although the patients who were involved in the study were healthier than others, still, some problems regarding therapy and adherence were detected. This study shows that adherence is a parameter that simultaneously depends on multivariety factors. Unfortunately, it is not possible to separately investigate the impact and effect of each factor. It was concluded that patients are often critical to themselves and admit that drug therapy is terminated due to specific factors (e.g., forgetfulness). In the future, it will be important to determine the causes of an already known problem, rather than investigate the factors of adherence. It was found that patients do not always pass on important information about medicines and food supplements to their physician. As a result, healthcare professionals are misinformed, and the risk of errors is increased.

## 5. Conclusions

The prevalence of non-adherence was 46.20%. The following correlations were found:
The adherent patients rated themselves higher than did the non-adherent ones in relation to the extent to which they follow the medication prescriptions of their family physician.The adherence level correlated with the patients’ age and did not depend on gender—the older the patient, the more adherent he or she is. The difference between adherence and number of prescribed drugs, number of food supplements, education level, and BMI was not statistically significant. It is important to increase the sample size.A higher adherence level was observed for those patients who had been taking medications for hypertension for a longer period of time.Anxiety and doubt are possibly the factors that influence the patients’ medication-taking habits; these factors were also the cause for the patient not informing their family physician regarding other medications and nutritional supplements that they were taking.

## Figures and Tables

**Figure 1 medicina-55-00715-f001:**
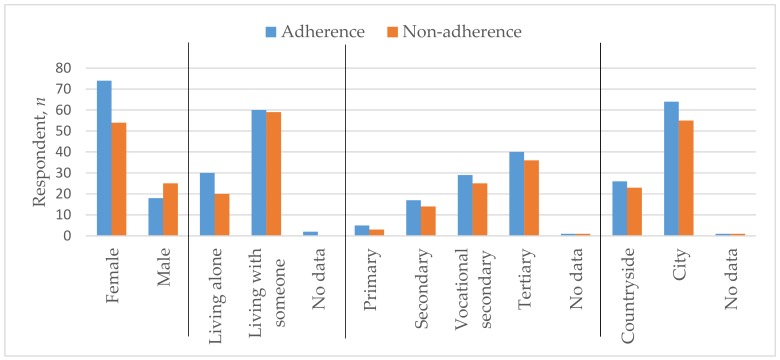
Sex, marital status, education level, and residence characteristics of adherent and non-adherent groups.

**Figure 2 medicina-55-00715-f002:**
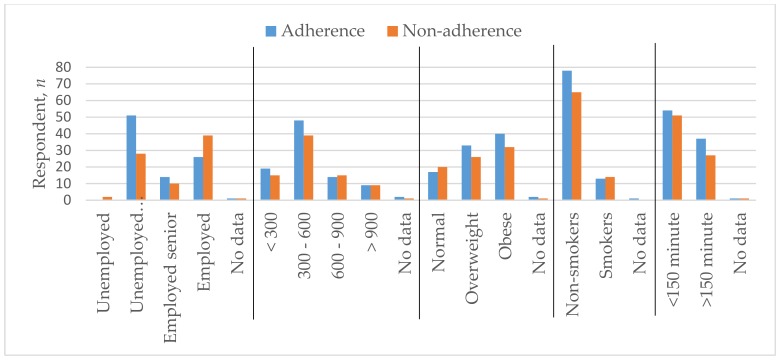
Employment status, monthly income after taxes in euros, BMI, smoking status, and physical activity characteristics of adherent and non-adherent groups.

**Figure 3 medicina-55-00715-f003:**
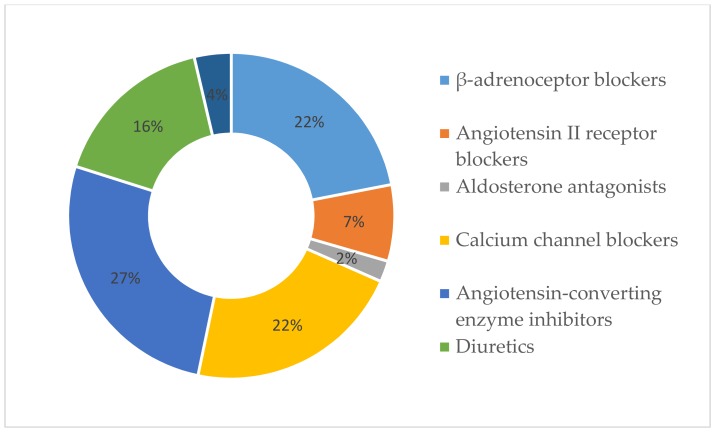
Percentages of used antihypertensive medications.

**Table 1 medicina-55-00715-t001:** Demographic characteristics of adherent and non-adherent groups.

Characteristic	Adherence		Non-Adherence	
	Mean	SD	Mean	SD
Age	67.73	12.27	60.52	13.63
	***n***	**Percent**	***n***	**Percent**
Sex				
Female	74	80.4	54	68.4
Male	18	19.6	25	31.6
Marital status				
Living alone	30	32.6	20	25.3
Living with someone	60	65.2	59	74.7
Education Level				
Primary	5	5.4	3	3.8
Secondary	17	18.5	14	17.7
Vocational secondary	29	31.5	25	31.6
Tertiary	40	43.5	36	45.6
Residence				
Countryside	26	28.3	23	29.1
City	64	69.6	55	69.6
Employment status				
Unemployed	0	0.0	2	2.5
Unemployed senior	51	55.4	28	35.4
Employed senior	14	15.2	10	12.7
Employed	26	28.3	39	49.4
Monthly income after taxes in euros				
<300	19	20.7	15	19.0
300–600	48	52.2	39	49.4
600–900	14	15.2	15	19.0
>900	9	9.8	9	11.4
Body mass index (BMI)				
Normal	17	18.5	20	25.3
Overweight	33	35.9	26	32.9
Obese	40	43.5	32	40.5
Smoking status				
Non-smoker	78	84.8	65	82.3
Smoker	13	14.1	14	17.7
Physical activity				
<150 min	54	58.7	51	64.6
>150 min	37	40.2	27	34.2

**Table 2 medicina-55-00715-t002:** Correlation between adherence level and other parameters.

Parameter	*p*-Value	Correlation Coefficient
Age	0.001	0.263
Daily medicine amount	0.124	0.119
Food supplement amount	0.551	0.047
Duration of disease	0.012	0.192
Following the medication prescriptions of their family physician	<0.0001	0.384
Education level	0.312	−0.780
BMI	0.095	0.129

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
