# Peer review of "Knowledge about Disease, Medication Therapy, and Related Medication Adherence Levels among Patients with Hypertension"

_medicina, 2019, doi:10.3390/medicina55110715_

Round 1

Reviewer 1 Report

This is a paper aimed to assess adherence of hypertensive patients according to several issues , including the  knowledge about disease and its control,  some demographic characteristics and the type of drugs administered .

The data were obtained in 171 out of 187 patients , attending to 12 pharmacies in Latvia, by the use of the  Modified 8-point Morisky Scale(MMAS-8).

The results show a prevalence of non adherence of 46 %, higher in patients of younger age , with a recent diagnosis of hypertension.

Major comments

The results confirm the association between non –adherence and some demographic and social characteristics of patients. The usefulness of the pharmacists intervention in administering the questionnaire should be proven in the follow up of the same patients .

It is not clear whether patients were free of previous CV disease or not .

Authors  found no influence of the number of drugs prescribed and non adherence . This is in contrast with what has been previously shown by several authors . Would authors comment about this discrepancy in the discussion ?

Minor points

 It is not clear what authors mean by the sentence “ The higher is the patient’s self-rating according to how much they follow the prescriptions, the higher is the level of adherence (p<0.0001, β = 0.38)” Can Authors better explain ?

English style is poor and needs to be improved

Author Response

Manuscript ID: medicina-545670

Dear Editor,

Please find below our response to the comments received as well as actions taken to accommodate changes to the manuscript.

Should you have any queries, please do not hesitate to contact us.

Major comments

The results confirm the association between non –adherence and some demographic and social characteristics of patients. The usefulness of the pharmacists intervention in administering the questionnaire should be proven in the follow up of the same patients .

We comply that in this study it is not possible to evaluate the usefulness of the pharmacist. Based on other studies, adherence to medicine can be improved by involving not only doctors but also pharmacists. In this study we did not assess the role of the pharmacist in increasing adherence in patients with hypertension; nevertheless, it could be taken into account in future studies.

It is not clear whether patients were free of previous CV disease or not.

The selection criteria were respondent age over 18 years, diagnosis of arterial hypertension and taking antihypertensive medicines for at least one year. Respondents may also have other cardiovascular disesase.

Authors  found no influence of the number of drugs prescribed and non adherence . This is in contrast with what has been previously shown by several authors . Would authors comment about this discrepancy in the discussion?

The results of other studies show that there is correlation between adherence and the number of prescribed medications. In this study difference between adherence and number of prescribed drugs was not statistically significant. It is important to increase the sample size.

Minor points

It is not clear what authors mean by the sentence “ The higher is the patient’s self-rating according to how much they follow the prescriptions, the higher is the level of adherence (p<0.0001, β = 0.38)” Can Authors better explain?

The higher the self-rated extent from 0 to 10 to which the patient takes their antihypertensives exactly as instructed by their physician, the higher the level of adherence (p < 0.0001, β = 0.38), where at "0", the patient does not follow physician instructions at all, and at "10", the patient completely follows the physician’s instructions. Non-adherent patients tend to assess their medication-taking behavior more critically than adherent patients.

English style is poor and needs to be improved.

We used a professional MDPI English editing service. 

Anna Gavrilova

Reviewer 2 Report

Overall, this article shows it has done the research to choose a valid methodology, an interesting aspect to analyze adherence through demographic associations, and reason for relevance in the pharmacy community. There is much to be done in editing the paper, as the grammar and syntax distract from the content and Table 1 is not very clear in what it is trying to depict. The following areas for improvement are noted:

ABSTRACT

Line 14 stating that “it is not possible to obtain higher therapeutic benefit” assumes it to be fact that therapeutic effectiveness cannot be changed and may distort what the authors are trying to say. This sentence could be altered to reflect that there is a limit to improving therapeutic benefit with the knowledge of patient behavioral barriers.

This is a significant study in a few ways. Although a lot has been done within the area of adherence, this study does unearth some issues (lack of awareness of goal, correlation between self report and actual adherence, age related adherence etc.) that are addressable and don't always show up in studies. A conceptual framework such as Theory of Planned Behavior or any health behavior model may help in aligning the findings. Maybe bring it up in discussion?

INTRODUCTION

In Line 43, when hypertension is defined, the guidelines or organization referred to for this definition should be stated in addition to the citation provided, since guidelines define hypertension differently.

Lines 39-41: It may be helpful to the introduction to give a bit of background on the different types of hypertension if arterial hypertension is looked at specifically.

Lines 51-52, to further strengthen the importance and weight of pharmaceutical treatment for hypertension, it could be helpful to expand upon “correct pharmacotherapy regimen.” What is the accepted regimen per which guideline being used in these countries or in the population being tested and what are some examples of them?

MATERIALS AND METHODS

Line 86: which kinds of pharmacies in Latvia were being used? Are they hospital related, independent, community, or all of the above?

How long did it take the patients to complete the questionnaire, and how was it administered? It may be beneficial in the 2.2 Questionnaire section to define who provided the questionnaires and how they were completed to prevent any questions about bias and to give more clarity on the method.

RESULTS

Is there a difference between the men and women? Is there an age difference in results? The demographics characteristics are great but only provide such, and are not referred to in their adherence.

There are a few times when open-ended questions seem to be referred to, i.e. line 167 with the frequently mentioned reasons for interrupting or terminating the therapy but these are not depicted anywhere in the figures/charts/diagrams or shown in either figure. How did a patient “admit it was difficult for them to understand what controlled blood pressure means?” (line 165). Was it through the survey or in the face to face talks to pharmacists? The direction of the results is getting lost with the description of the data.

The information in medication therapy overview (linen 170 and on) is not shown anywhere in any figures.

DISCUSSION

Well done but some more focus on the "so what" what kind of further research or interventions can arise from this research?

CONCLUSIONS

Line 282: It seemed to come out of nowhere that anxiety and doubt were the main factors that influenced medication taking habits yet they were not included in the demographics. What happened?

FIGURES/CHARTS/DIAGRAMS

Considering the medication therapy overview, there is no figure to show the results of the survey aside from the p-value and correlation coefficient in table 2.

Table 1 is also a bit confusing using the mean next to the percent for the lines starting with marital status since those two values aren’t related. Are the percent values supposed to add up to 100%? The table is extremely confusing. I do not understand what it means. How can there be a mean number of females and males? What does that add? Or is this not labeled properly?

Maybe bar graphs could illustrate the results in a more clear way? 

Author Response

Manuscript ID: medicina-545670

Dear Editor,

Please find below our response to the comments received as well as actions taken to accommodate changes to the manuscript.

Should you have any queries, please do not hesitate to contact us.

ABSTRACT

Line 14 stating that “it is not possible to obtain higher therapeutic benefit” assumes it to be fact that therapeutic effectiveness cannot be changed and may distort what the authors are trying to say. This sentence could be altered to reflect that there is a limit to improving therapeutic benefit with the knowledge of patient behavioral barriers.

We have included this change.

This is a significant study in a few ways. Although a lot has been done within the area of adherence, this study does unearth some issues (lack of awareness of goal, correlation between self report and actual adherence, age related adherence etc.) that are addressable and don't always show up in studies. A conceptual framework such as Theory of Planned Behavior or any health behavior model may help in aligning the findings. Maybe bring it up in discussion?

Your comments suggestions was greatly appreciated, thus we have included in discussion part: A variety of determinants, both rational and emotional, that influence health behavior have been described. Adherence to pharmaceutical and non-pharmaceutical treatment is one of these behaviors. Using a conceptual framework such as the Theory of Planned Behavior may help provide deeper understanding of the findings of this research and could provide for further research initiatives.

INTRODUCTION

In Line 43, when hypertension is defined, the guidelines or organization referred to for this definition should be stated in addition to the citation provided, since guidelines define hypertension differently.

We have included this change.

Lines 39-41: It may be helpful to the introduction to give a bit of background on the different types of hypertension if arterial hypertension is looked at specifically.

Arterial hypertension was chosen because it is a serious problem. According to WHO data it is major indicators in Europe (as well as in Latvia). Description of other types of hypertension types such as pulmonary or cranial would be a deviation from the main topic. A more detailed description of other hypertension types would be particularly relevant, however the major problem in cardiology is AH.

Lines 51-52, to further strengthen the importance and weight of pharmaceutical treatment for hypertension, it could be helpful to expand upon “correct pharmacotherapy regimen.” What is the accepted regimen per which guideline being used in these countries or in the population being tested and what are some examples of them?

We have included this change. We cannot impugn drug therapy which is chosen by the physician.

MATERIALS AND METHODS

Line 86: which kinds of pharmacies in Latvia were being used? Are they hospital related, independent, community, or all of the above?

We have included this change.

How long did it take the patients to complete the questionnaire, and how was it administered? It may be beneficial in the 2.2 Questionnaire section to define who provided the questionnaires and how they were completed to prevent any questions about bias and to give more clarity on the method.

We have included this change.

RESULTS

Is there a difference between the men and women? Is there an age difference in results? The demographics characteristics are great but only provide such, and are not referred to in their adherence.

The adherence level correlated with the patients’ age and did not depend on gender—the older the patient, the more adherent he or she is. The difference between adherence and number of prescribed drugs, number of food supplements, education level, and BMI was not statistically significant. It is important to increase the sample size. We are planning to continue this study to find more statistically significant correlation.

There are a few times when open-ended questions seem to be referred to, i.e. line 167 with the frequently mentioned reasons for interrupting or terminating the therapy but these are not depicted anywhere in the figures/charts/diagrams or shown in either figure.

The reasons for termination drug therapy cannot be represented correctly using charts for data vizualization, due to patients were able to indicate multiple answers. This manuscript covers the three main reasons for termination of the drug therapy: forgetfulness, concerns about getting addicted to the medications and undesirable side effects.

How did a patient “admit it was difficult for them to understand what controlled blood pressure means?” (line 165). Was it through the survey or in the face to face talks to pharmacists? The direction of the results is getting lost with the description of the data.

We have included this change.

The information in medication therapy overview (linen 170 and on) is not shown anywhere in any figures.

We have included this change.

DISCUSSION

Well done but some more focus on the "so what" what kind of further research or interventions can arise from this research?

We have included this change.

CONCLUSIONS

Line 282: It seemed to come out of nowhere that anxiety and doubt were the main factors that influenced medication taking habits yet they were not included in the demographics. What happened?

Anxiety and doubt are possibly the factors that influence the patients’ medication-taking habits; these factors were also the cause for the patient not informing their family physician about other medications and nutritional supplements they were taking.

FIGURES/CHARTS/DIAGRAMS

Considering the medication therapy overview, there is no figure to show the results of the survey aside from the p-value and correlation coefficient in table 2.

The medication therapy overview is a descriptive data that describes the most commonly selected medication at the moment.

Table 1 is also a bit confusing using the mean next to the percent for the lines starting with marital status since those two values aren’t related. Are the percent values supposed to add up to 100%? The table is extremely confusing. I do not understand what it means. How can there be a mean number of females and males? What does that add? Or is this not labeled properly?

Maybe bar graphs could illustrate the results in a more clear way?

Some respondents didn’t answer the questions correctly, therefore these data are presented as missing data and they do not affect other data. Data visualization has been performed by creating two charts. It is shown why in table is no 100 percent in total.

Anna Gavrilova

Round 2

Reviewer 1 Report

authors have addressed all major points raised by this reviewer

Author Response

Thank you for the information.

Reviewer 2 Report

The authors seem to have addressed all reviewer comments, except the conceptual framework needed to define their project.

Author Response

This study contains only descriptive statistic and we did not perform any interventions to estimate effect of the variables. We agree that this is important to explain patient behavior regarding to adherence. Based on the reviewer's recommendation, we have supplemented discussion part by adding following text:

A variety of determinants, both rational and emotional, that influence health behavior have been described. Adherence to pharmaceutical and non-pharmaceutical treatment is one of these behaviors. Using a conceptual framework such as the Theory of Planned Behavior may help provide deeper understanding of the findings of this research and could provide for further research initiatives.